# Preparation and Optimization of MiR-375 Nano-Vector Using Two Novel Chitosan-Coated Nano-Structured Lipid Carriers as Gene Therapy for Hepatocellular Carcinoma

**DOI:** 10.3390/pharmaceutics16040494

**Published:** 2024-04-03

**Authors:** Bangly Soliman, Ming Ming Wen, Eman Kandil, Basma El-Agamy, Amira M. Gamal-Eldeen, Mahmoud ElHefnawi

**Affiliations:** 1Department of Biochemistry, Faculty of Science, Ain Shams University, Cairo 11566, Egypt; beny_12788@yahoo.com (B.S.);; 2Biomedical Informatics and Chemo-Informatics Group, Informatics and Systems Department, National Research Centre, Cairo 12622, Egypt; 3Faculty of Pharmacy, Pharos University, Alexandria 21648, Egypt; 4Clinical Laboratory Sciences Department, College of Applied Medical Sciences, Taif University, P.O. Box 11099, Taif 21944, Saudi Arabia; amabdulaziz@tu.edu.sa

**Keywords:** hepatocellular carcinoma, miR-375, nanostructured solid liquid lipid carrier, lyophilization, transfection efficiency, stability, cytotoxicity

## Abstract

Currently, there is still a lack of effective carriers with minimal side effects to deliver therapeutic miRNA. Thus, it is crucial to optimize novel drug delivery systems. MiR-375 has proven superior therapeutic potency in Hepatocellular carcinoma (HCC). The purpose of this study was to fabricate 2 novel and smart nano-carriers for the transportation efficiency of miR-375 in HCC cells and enhance its anti-tumor effects. We established the miR-375 construct through the pEGP- miR expression vector. Two nano-carriers of solid/liquid lipids and chitosan (CS) were strategically selected, prepared by high-speed homogenization, and optimized by varying nano-formulation factors. Thus, the two best nano-formulations were designated as F1 (0.5% CS) and F2 (1.5% CS) and were evaluated for miR-375 conjugation efficiency by gel electrophoresis and nanodrop assessment. Then, physio-chemical characteristics and stability tests for the miR-375 nano-plexes were all studied. Next, its efficiencies as replacement therapy in HepG2 cells have been assessed by fluorescence microscopy, flow cytometry, and cytotoxicity assay. The obtained data showed that two cationic nanostructured solid/liquid lipid carriers (NSLCs); F1 and F2 typically had the best physio-chemical parameters and long-term stability. Moreover, both F1 and F2 could form nano-plexes with the anionic miR-375 construct at weight ratios 250/1 and 50/1 via electrostatic interactions. In addition, these nano-plexes exhibited physical stability after three months and protected miR-375 from degradation in the presence of 50% fetal bovine serum (FBS). Furthermore, both nano-plexes could simultaneously deliver miR-375 into HepG2 cells and they ensure miR re-expression even in the presence of 50% FBS compared to free miR-375 (*p*-value < 0.001). Moreover, both F1 and F2 alone significantly exhibited minimal cytotoxicity in treated cells. In contrast, the nano-plexes significantly inhibited cell growth compared to free miR-375 or doxorubicin (DOX), respectively. More importantly, F2/miR-375 nano-plex exhibited more anti-proliferative activity in treated cells although its IC50 value was 55 times lower than DOX (*p*-value < 0.001). Collectively, our findings clearly emphasized the multifunctionality of the two CS-coated NSLCs in terms of their enhanced biocompatibility, biostability, conjugation, and transfection efficiency of therapeutic miR-375. Therefore, the NSLCs/miR-375 nano-plexes could serve as a novel and promising therapeutic strategy for HCC.

## 1. Introduction

Hepatocellular carcinoma (HCC) is a prevalent malignancy, ranking fifth among all cancers and second in cancer-related deaths globally. It accounts for 90% of all primary liver tumors, highlighting its significance [1]. Current treatment strategies for HCC include chemotherapy, radiotherapy, and immunotherapy; however, these methods have limited efficacy, high toxicity, and lack tumor selectivity [2,3]. The 5-year relative survival rate for all SEER stages combined is only about 20%, indicating a poor prognosis [4]. Therefore, there is an urgent need for the development of more effective therapeutic approaches to achieve better tumor control.

MicroRNAs (miRNAs) have been extensively researched as potential therapeutic agents to combat various tumors due to their ability to act as master regulators of gene expression at both the post-transcriptional and translational levels [5,6]. Each miRNA can perfectly base pair with tens to hundreds of mRNA targets, directly repressing their pathways [7]. Nevertheless, miR dysregulation plays a fundamental role in tumor initiation and progression [8]. Modulating the function of miRNAs, either by restoring their expression or inhibiting their overexpression, can influence their regulatory network and potentially halt cancer growth [9,10]. 

MiR-375 is a conserved noncoding RNA that is frequently downregulated in various tumors. Its upregulation has been shown to inhibit malignant traits of cancer cells, making it a potent onco-suppressor and one of the core down-regulated miRs in HCC [11,12,13,14,15,16,17,18,19,20]. The hypermethylation of CpG islands of miR-375 promoter has been reported as the main cause of its down-regulation [21,22]. Restoring the expression of miR-375 has been found to significantly repress the major hallmarks and all signaling networks of HCC, including cell growth, proliferation, anti-apoptosis, angiogenesis, glucose metabolism, autophagy, drug resistance, migration, and invasion [23,24,25,26,27,28,29,30]. These suppressive effects of miR-375 are achieved via its multi-targeting key oncogenes involved in hepatocarcinogenic pathways. These findings potentially suggest that miR-375 may be a promising therapeutic strategy for HCC.

Efficiently delivering functional miRNA into target cancer cells remains a significant challenge due to its negative charge, limited penetration, short half-life, and susceptibility to degradation by nucleases [31,32]. To overcome these critical obstacles, various viral and non-viral delivery systems have been developed [33,34]. Non-viral-based carriers, such as nanoparticles (NPs), have shown remarkable potential for effective miRNA-based therapeutics [18,19,35,36]. There are several types of NPs available, including lipidic nanoparticles, polysaccharides, and others. These non-viral delivery systems offer several advantages in miRNA delivery [37,38,39]. 

Lipid-based nanoparticles (LBNPs) have shown remarkable potential for miRNA delivery due to their ease of production, efficient encapsulation and stabilization of the miRNA payload, and ability to facilitate tumor-specific delivery, high intracellular uptake, and enhanced bioavailability with minimal toxicity [40]. In line with these, cancer cells are known to contain a high percentage of lipids, which are characterized as landmarks of tumor aggressiveness [41,42], Several lipid-based nano-carriers for miRNA delivery are well-absorbed into target cells, demonstrating efficient abilities. Examples of these carriers include nanostructured lipid carriers (NSLCs) [43], solid lipid nanocarriers (SLNs) [44], high-density lipoproteins [45], stable nucleic acid-lipid particles [46] and cationic liposomes [47].

Nanostructured lipid carriers (NSLCs) are considered the second generation of LBNPs which are made by combining solid and liquid lipids [48]. The addition of liquid lipids transforms the perfectly crystalline structure of SLNs into an imperfect, amorphous, and less organized matrix [49]. This lipid matrix provides more space for drug loading, enhances encapsulation and stability, and modulates targeted delivery [50,51,52,53,54,55,56]. NSLCs have been developed as novel carriers for the effective delivery of multiple miRNAs in tumor gene therapy. Examples of these miRNAs include miRNA-125-a-5p, anti-miR-221, miR-34a, let-7-a, and anti-miR-21 [57,58,59,60,61]. On the other hand, chitosan (CS) is a cationic polysaccharide that has been widely used in the synthesis of nanocarriers for drug delivery, including miRNA delivery [62,63,64]. It is highly biocompatible, biostable and target-specific, with a strong binding affinity for miRNA [65,66]. In addition, it is characterized as a non-toxic and a biodegradable delivery vehicle [67]. Mechanistically, CS amino groups are ionized in acidic environments, allowing them to interact with other molecules through the formation of electrostatic complexes or multilayer structures [68,69]. For instance, CS has been reported to form a nano-complex with hyaluronan which efficiently encapsulated vinblastine sulfate, a potential cytotoxic drug against myeloid leukemia. This optimized cargo demonstrated active targeting and sustained therapeutic effects in vitro [70]. More importantly, CS NP has been shown to form stable complexes with miR-217 [71], and miR-122 [72] in the treatment of HCC, as well as with miR-200c [73,74], and miR-34a [75,76] in the treatment of breast cancer.

Despite the progress made in developing vectors for miRNA delivery to cancer cells, there is still a need for further development and optimization of ideal vectors for delivering miRNA to target cells to solve the shortage of the currently available ones and dramatically improve therapeutic outcomes. In this study, the aim was to develop a smart gene therapy based on a miR-375 vector as a therapeutic strategy for HepG2 cells. We fabricated two novel nanocarriers which consisted of chitosan-coated nanostructured lipid carriers. The nanocarriers were optimized to possess several functionalities, such as high biocompatibility, stability, and cell-specific targeting, while also reducing potential toxicities. By enhancing the re-expression of miR-375 and its tumor-suppressive roles, these nanocarriers can effectively inhibit the malignant phenotypes of cancer cells.

## 2. Materials and Methods

### 2.1. Materials

Chitosan (medium MW, 75–85% deacetylated), oleic acid, Kallichore P188 (Poly (ethylene glycol)-block-poly (propylene glycol)-block-poly (ethylene glycol), Avicel (microcrystalline cellulose), and trehalose were purchased from Sigma-Aldrich Co. (St. Louis, MO, USA). Precirol^®^ ATO 5 (Glyceryl Di stearate NF/Glyceryl palmitostearate), and Gelucire 50/13 (Stearoyl polyoxylglycerides NF/Stearoyl macrogol glycerides EP) were kind gifts from Gattefosse (Lyon, France). Mannitol, lactose, and sucrose (El-Gomhoreya, Cairo, Egypt), Dulbecco’s Modified Eagle’s medium (DMEM, Gibco Invitrogen, Darmstadt, Germany), fetal bovine serum (FBS), Trypsin EDTA, penicillin and streptomycin (Biowest, Lakewood Ranch, FL, USA). DNaeay DNA extraction kit, Hot star taq PCR master mix (Qiagen, Hilden, Germany). pEGP miR-cloning and expression vector (Cell Biolabs, San Diego, CA, USA), fast digest BamHI and fast digest NheI enzymes, T4 ligase, Gel extraction kit, Plasmid miniprep, Turbofect transfection reagent (Thermo Scientific, Waltham, MA, USA). HI pure endotoxin-free plasmid maxiprep (Invitrogen, Waltham, MA, USA), MTT reagent (Sigma Aldrich, St. Louis, MO, USA). All other chemicals used were of pharmaceutical grade or the highest commercially available grade.

### 2.2. Generation of Stable miR-375 Expression Construct

To prepare the HCC suppressor miR-375 for use in this study, commercially provided *E. coli* cells containing pEGP miR-cloning and expression vector with a Green Fluorescent Protein (GFP) selection marker were cultured on 1% LB Agar supplemented with ampicillin at 37 °C for 32 h. A single colony was then picked and grown in 10 mL of 1% LB media supplemented with ampicillin at 37 °C for 24 h. The vector was subsequently purified from the liquid culture using a Plasmid Miniprep kit following the manufacturer’s protocol, and the concentration of the pEGP miR vector was determined by Nanodrop at 260 nm. Double restriction digestion of the vector was carried out using fast-digest BamHI and fast-digest NheI enzymes. The HCC suppressor miR-375 was designed and prepared based on our previous work [17,18,19,20]. Total genomic DNA was isolated and the pre-miR-375 sequence was amplified using designed forward 5′ CGGACCTGAGCGTTTTGTTC 3′ and reverse 5′ TACGGTTGAGATGGCGGTG 3′, and the pure miR-375 sequence was cloned into the pEGP miR-cloning and expression vector. The recombinant miR-375 expression construct was verified by Sanger sequencing using specific forward 5′TTTGCACCATTCTAAAGAAT3’ and reverse 5′AAACCTCTTACATCAGTTAC3′ sequencing primers.

### 2.3. Preparation and Optimization of Chitosan-Coated NSLCs

The NSLC was prepared using high-speed homogenization and contained Precirol ATO-5, oleic acid, Gelucire, and Kolliphore P188 which were chosen for their regulatory status (GRAS, USA, FDA), purity, chemical stability, ability to enhance the cationic nano-formulation of miR-375, and biodegradability of these carriers after miR delivery. Both lipid and aqueous phases were first heated separately to 60 °C on magnetic stir plates and then mixed under high-speed homogenizing (IKA T25, Digital ULTRA TURRAX^®^, IKA, Staufen, Germany) at 20,000 rpm for 10 min, followed by 5 min sonication. Chitosan solution was used as a final coat for the NSLC. We prepared two concentrations of CS, 0.5% and 1.5%, and pH was adjusted to 4–5.5 by 1 N NaOH. An equal volume of either 0.5% or 1.5% of CS was added dropwise to the mixture. The formed nano-emulsions were stirred continuously at 100 rpm overnight at room temperature to achieve 2 positively charged delivery systems designated as F1 and F2. The nano-formulations were examined by various factors such as solid: liquid lipid ratio, surfactant concentration, CS concentration, and CS pH to investigate their impact on PS, ZP, and PDI (Table 1).

### 2.4. Nano-Formulation of miR-375 

MiR-375 was loaded into CS-coated NSLC using the self-assembly method. First, stock solutions of 4 mg/mL cationic NSLC colloids were prepared and filtered with a 0.45 mM filter to increase their uniformity which were then mixed with plasmid carrying miR-375 at different weight ratios, vortexed for 2–3 min, and incubated for 30 min at room temperature to allow the formation of nano-electrostatic complexes.

### 2.5. Evaluation of Physicochemical Characteristics

#### 2.5.1. Particle Size (PS), Zeta Potential (ZP), and Polydispersity Index (PDI)

The studied samples were diluted with distilled water at a final concentration of 0.006 mg/mL to prevent multiple scattering. Then, they analyzed using dynamic light scattering with a Zeta-sizer Nano ZS instrument (Malvern Instruments, Worcestershire, UK) at a scattering angle of 173° and a temperature of 25 °C according to the manufacturer’s recommendation [77]. Each measurement was repeated three times. 

#### 2.5.2. Morphological Evaluations

The shape and surface morphology of blank F1, F2, and F1/miR-375, F2/miR-375, were examined by Transmission electron microscope (TEM) (Joel JEM 1230, Tokyo, Japan). The diluted samples were negatively stained with a 2% aqueous solution of sodium phosphotungstate for 5 min and placed on a copper grid for 10 min drying at 25 °C.

#### 2.5.3. Fourier Transform Infrared Spectrophotometry(FT-IR)

To confirm the formation of the nano-structure and investigate possible interactions between the miR drug and nano-formulation components, FT-IR spectra of the excipients, naked miR-375, blank F1, blank F2, F1/miR-375, and F2/miR-375 were recorded. Absorption peaks were measured in the region between 450 and 4000 cm^−1^ using a Cary 630 FT-IR Spectrometer (Agilent, Santa Clara, CA, USA). All samples were analyzed in their original forms, while miR-375 was measured in the buffer for miRNA.

### 2.6. Gel Retardation Assay

The optimal capability of cationic F1 and F2 to conjugate miR-375 was evaluated using agarose gel (2%, *w*/*v*) electrophoresis with ethidium bromide. The electrophoresis was performed under a current voltage of 120 V for 25 min in a Tris-acetate (TAE) running buffer. F1/miR-375 and F2/miR375 nano-vectors were prepared at different weight ratios by varying the concentration of F1 and F2 (1–350 μg) while maintaining a fixed concentration of miR-375 (1 μg) in all ratios. The conjugated nano-plexes were then incubated at 25 °C for 30 min, followed by combining 15 µL of each suspension with 2.5 µL of loading buffer (Biolabs, Hitchin, UK). Images were obtained using an ultraviolet transilluminator and a digital Imaging system (GL 200; Kodak, Windsor, CO, USA), and the results were analyzed.

### 2.7. MiR-375 Loading Efficiency

The amount of miR-375 loaded into the nano-formulations was determined using spectrophotometry. The nano-plexes were centrifuged at 10,000 rpm at 25 °C, and the concentration of unconjugated miR-375 in the supernatant was analyzed using a Nanodrop^®^ ND1000 spectrophotometer (Thermo Scientific, USA) at 260 nm. The conjugation efficiency (CE) (%) was calculated using a formula previously reported [78]:(Total weight used of DNA − weight of free DNA)/total weight of DNA × 100

### 2.8. Lyophilization Study

The two nano-plexes F1/miR-375 and F2/miR-375 were freeze-dried to evaluate the optimal state for their long-term stability. Cryoprotectants are often added before lyophilization to produce a stable unimodal size distribution during the stressful lyophilization process and the reconstitution of lyophilized formulations [79,80]. We evaluated several promising cryoprotectants, including Avicel, mannitol, lactose, trehalose, and sucrose, to determine their impact on the aggregation of the nano-plexes. Before the freeze-drying process, phosphate-buffered saline (PBS) containing 4% (*w*/*v*) cryoprotectant was added to freshly prepared F1/miR-375 and F2/miR-375. The samples were then frozen at −80 °C for 24 h. and transferred to a freeze dryer under vacuum (Human Lab Instrument Co., Seoul, Republic of Korea).

To reconstitute the freeze-dried powder, 600 µL of deionized water was added to 30 mg of the powder, except for samples containing Avicel, which were re-dispersed using 5% *w*/*v* sodium hydroxide. Finally, all reconstituted samples were subjected to PS and PDI analysis, and compared to the frozen samples in a freezer at −20 °C without the addition of cryoprotectants. Each sample batch was prepared in triplicate.

### 2.9. Stability Studies

#### 2.9.1. Storage (Physical) Stability Study

The stability of the F1/miR375 and F2/miR375 to preserve miR-375 against leakage was monitored for 3 months and evaluated using DLS analysis. The frozen at −20 °C samples were equilibrated to room temperature before the measurement. The lyophilized samples with sucrose were reconstituted by adding nuclease-free water, gently mixed, and then measured. Freshly prepared F1/miR-375 and F2/miR-375 colloids were used as controls. All measurements were performed in triplicate.

#### 2.9.2. Serum Stability Study

The protective effect and stability of the nano-formulations against degradation by endonucleases were investigated in the presence of fetal bovine serum (FBS). Each miR-375 nano-vector, with a weight ratio of 250/1 for F1 and 50/1 for F2, was mixed with an equal volume of 50% *v*/*v* FBS. The mixtures were then incubated at 37 °C for 24 h and 1 week. The final samples were loaded onto a 1% agarose gel and subjected to electrophoresis at 120 V for 25 min and observed with an ultraviolet transilluminator and a digital imaging system. Furthermore, the final samples were tested in HepG2 cells for the presence of green fluorescence as detailed in the transfection efficiency section.

### 2.10. HCC Cell Culture

HepG2 cells which attained the major characteristics of HCC, were obtained from the ATCC bank. It was used as an in vitro model to study the therapeutic effect of the miR-375 nano-formulations. The cells were cultured in DMEM medium containing 10% (*v*/*v*) heat-inactivated FBS, as well as penicillin (100 U/mL) and streptomycin (100 U/mL, and maintained in a humidified incubator at 37 °C with 5% CO_2_. After three successive passages, the cells were seeded at a density of 10,000 cells/well in 200 µL of growth medium on a 96-well plate and at 80,000 cells/well in 600 µL of growth medium on 8 well slides for fluorescence microscopy. For flow cytometry, the cells were seeded at a density of 1 × 10^5^ cells/well in 1000 µL of growth medium in a 24-well plate. After 24 h, the culture medium was replaced with fresh media. This research was approved with the corresponding ethical approval code ASU-SCI/BIOC/2023/3/1 by the ethical committee at the Faculty of Science, Ain Shams University, Cairo, Egypt.

### 2.11. Transfection Efficiency and Cellular Uptake

#### 2.11.1. Qualitative Assessment by Fluorescence Microscopy

The efficacy of cell transfection and the intracellular release of miR-375 from the nano-formulations were examined using HepG2 cells. The cells were transfected with a naked miR-375 vector (1 µg) using a transfection agent Turbofect, F1/miR-375, and F2/miR-375 nano-plexes at weight ratios of (250/1 and 50/1) and incubated for 48 h. Afterward, the cells were rinsed twice with 600 µL PBS (pH 7.4) and fixed with 600 µL ice-cold methanol. The resulting images were examined under a fluorescence microscope (Olympus, Tokyo, Japan) to assess the green fluorescence of the co-expressed GFP marker at excitation and emission wavelengths of 395 nm and 509 nm, respectively.

#### 2.11.2. Quantitative Assessment by Flow Cytometry

Flow cytometry was employed to quantify the mean fluorescence intensity (MFI). Following 48 h of transfection, with and without pre-treatment with 50% FBS, cells were harvested using 150 μL of 0.05% trypsin, washed, and suspended in PBS. The cell suspension was then analyzed on the FL1 channel for green fluorescence using a fluorescence-activated cell sorter (FACS) to identify stable miR-375-expressing transfectants (BD Biosciences, San Jose, CA, USA). The MFI estimated the results compared to the control.

Remarkably, the serum stability of the F1/miR-375 and F2/miR-375 nano-plexes was assessed as follows: 

HepG2 cells were transfected with miR-375, F1/miR-375, and F2/miR-375 nano-plexes in the presence and absence of 50% FBS pre-treatment and incubated for 48 h at 37 °C and detected by fluorescence microscopy and flow cytometry as mentioned before.

### 2.12. Cytotoxicity Assessment

The in vitro anti-HCC activities of miR-375 nano-plexes were evaluated using the relative cell growth ratio measured by the MTT assay. On the day of transfection, the medium in untreated cells, which served as a blank control, was replaced with a fresh serum-free DMEM medium. Other groups were treated with fresh medium containing different concentrations of blank F1, blank F2 (10–500 µg/mL), 200 ng of naked miR-375, 4 µL of TurboFect transfection reagent (positive control), F1/miR-375 and F2/mir-375 nano-plexes, respectively. Doxorubicin (DOX) at concentrations of 200 ng (the same used for miR-375) and 11 µg, at which half maximal inhibitory concentration (IC50) is reached, was considerably used to provide a reasonable comparison [81]. After incubation for 48 h, all cells were washed with 200 µL PBS and replaced with 200 µL of growth medium containing 20 µL of 5 mg/mL MTT in each well. This was followed by an additional 4 h incubation to allow for the formation of formazan crystals. After the incubation, the solution in each well was removed, and the crystals were solubilized using 200 µL of DMSO. The absorbance was measured at 570 nm using a microplate ELISA reader (Biotech Instruments Inc., Winooski, VT, USA). The percentage of the tumor cell inhibition rate was calculated using the following formula:Cell viability (%) = A_570_ (test)/(A_570_ (control) × 100
Tumor cells inhibition rate (%) = 100% − cell viability (%)

A_570_ (test) represents a measurement from the well treated with F1, F2, F1/miR-375, and F2/miR-375, whereas A_570_ (control) represents a measurement from the well of control without treatment.

### 2.13. Statistical Analysis

All results were presented as mean ± standard deviation based on three repeated experiments. Statistical analysis of the different groups was conducted using GraphPad Prism version 8.0.0 for Windows (GraphPad Software, San Diego, CA, USA), and the one-way ANOVA test or t-test was performed accordingly. A *p*-value less than 0.05 was considered statistically significant, while a *p*-value less than 0.01 was considered very significant.

## 3. Results

### 3.1. HCC Suppressor miR-375 Expression Construct

The purified pre-miR-375 sequence (127 b.p) was successfully cloned in pEGP-miR cloning and expression vector with approximate size 5000 b.p (Figure 1). The recombinant miR-375 expression construct was verified by Sanger sequencing.

### 3.2. Factors That Influence PS, ZP, and PDI of the Nano-Formulations

The two nanoemulsions forming CS-coated NSLCs comprised solid/liquid lipids core alleviated by interfacial surfactant layers and CS coat. We carefully investigated the impact of various preparation factors of CS-coated NSLCs as carriers for miR-375 replacement therapy in HCC. These factors included the ratio of solid to liquid lipids, the concentration of surfactant, and the concentration and the pH of CS, as listed in Table 2. Precirol ATO-5 and oleic acid were used as the solid and liquid lipids, respectively to create a stable nano-structured colloid. There was a highly significant impact on the nano-lipid size by varying their included ratio in the formulation (*p* < 0.001). Meanwhile, formula-C (1:1) was selected for further investigation of other factors due to its smallest PS, strongest ZP, and modest PDI among all.

To prevent agglomeration and reduce the surface energy of nanolipids, we used Gelucire 50/13 and Kolliphore P188 as surfactants in a fixed ratio of 1:2. These surfactants have different Hydrophile-Lipophile Balance (HLB) values: 11 for Gelucire 50/13 and 29 for Kolliphore P188. Gelucire 50/13 contains both hydrophobic and hydrophilic components, such as PEG mono- and diesters with palmitic (C16) and stearic (C18) acids, glycerides (20%), and PEG esters (80%). Kolliphore^®^ P 188 is a non-ionic amphiphilic copolymer with hydrophobic polyoxypropylene and hydrophilic polyoxyethylene. We observed a significant decrease in PS with higher surfactant concentration (one-way ANOVA, *p* < 0.001). Therefore, formula-J with 3% surfactants was selected to continue the investigation of other factors because it had the smallest PS and PDI. Interestingly, decorating NSLC with different concentrations of CS had no significant influence on PS but showed a strong positive surface charge (*p* < 0.05). Formula-K was then selected for the pH study, and the result showed that the ZP of the NP decreased significantly from 28.03 ± 2.33 mV to 9.28 ± 4.12 mV when the pH of the CS solution increased from 4 to 5.5 by the addition of 1N NaOH (*p* < 0.001). Finally, we selected formula-P with 0.5% CS and formula-M with 1.5% CS for further overall study based on their modest PS and strongest ZP, and they represented the optimized cationic NSLCs which were labeled as F1 and F2, respectively.

### 3.3. Nano-Formulation of miR-375

The cationic F1 and F2 could be easily loaded with anionic double-stranded miR-375 to form two electrostatic nano-plexes bearing miR-375; F1/miR-375 and F2/miR-375.

### 3.4. Results of Physicochemical Characterization

#### 3.4.1. DLS Results

Table 3 summarizes the PS, ZP, and PDI of F1/miR-375 and F2/miR-375 compared to F1 and F2 blank formulations. Results showed that the PS of F1/miR-375 and F2/miR-375 significantly increased compared to blank formulations due to miR-375 conjugation on the surface of cationic NSLCs (*p* < 0.001). Meanwhile, the ZP of F1/miR-375 decreased to 7.7 ± 1.02 mV compared to 32.5 ± 3.86 mV for the blank F1, and the ZP of F2/miR-375 decreased to 5.8 ± 1.34 mV compared to 61.8 ± 2.74 mV for the blank F2, indicating that miR-375 was well conjugated on the surface of both F1 and F2 due to the negatively charged miR-375. Additionally, there was a slight change in the PDI of nanoplexes compared to blank F1 and F2. According to this overall data, blank F2 represents the optimal formula to nano-formulate miR-375 due to its increasing CS concentration which produced more positive charge on its surface.

#### 3.4.2. TEM Results

Morphology images of blank F1, blank F2, F1/miR375, and F2/miR375 were investigated by TEM as shown in Figure 2. The images indicated that the PS of the NPs was consistent with the measurements obtained by Zeta-sizer. For instance, all the NSLCs exhibited spherical shapes and miR-375 was observed to be conjugated on their surfaces.

#### 3.4.3. FT-IR Results

Analysis of FT-IR spectra showed clearly the excipients were well integrated into the two nano-formulations, F1 and F2. However, there is no interaction between the miR-375 drug and F1 or F2 (Figure 3). Oleic acid, an unsaturated fatty acid, exhibited a characteristic peak at 1707.2 cm^−1^, corresponding to the C=O stretching vibration of the carboxylic acid group. Kolliphore 188, a non-ionic surfactant and block copolymer of ethylene oxide and propylene oxide, showed characteristic peaks at 2876.2 cm^−1^, corresponding to the stretching vibration of the CH_2_ groups in the molecule, at 1465.9 cm^−1^ for the bending vibration of the CH_2_ groups, and several peaks around 1100–1000 cm^−1^, corresponding to the stretching vibrations of the C-O and C-C bonds. Gelucire, a mixture of mono-, di-, and triglycerides of fatty acids, displayed peaks around 2920–2850 cm^−1^ for the stretching vibration of the CH_2_ groups in the fatty acid chains. A peak at 1733.5 cm^−1^ corresponded to the C=O stretching vibration of the ester groups, and a peak at 1102.7 cm^−1^ corresponded to the C-O-C stretching vibration of the glycerol backbone. Furthermore, CS is a natural poly (amino saccharide) derived from the deacetylation of chitin, consisting predominantly of unbranched chains of (1→4)-2-acetoamido-2-deoxy-d-glucose. The previously reported spectra of CS showed characteristic peaks at 1418.4 cm^−1^ related to the -NH bending vibration, at 3362 cm^−1^ for the intramolecular hydrogen bonds of O-H stretching vibration, and at 1645.8 cm^−1^ indicating the presence of C=O stretching of amide I [82]. Consistently, both F1 and F2 exhibited similar bands to CS, indicating the coat of CS on their surfaces. In addition, the absorption peaks of F1/miR-375 and F2/miR-375 in blue at 3400 cm^−1^ and 3200 cm^−1^ showed no significant interference with each nano-formulation, indicating the maintained structure after conjugation.

### 3.5. Optimization of miR-375 Loaded into the CS-Coated NSLCs

The conjugation efficiency of the two CS-coated NSLC/miR-375 was investigated by electrophoretic mobility. Gradual miRNA retardation was observed when the weight ratio (μg/μg) was lower than 250/1 for F1/miR-375 and lower than 50/1 for F2/miR-375 nano-vectors. When the weight ratio was 250 or above for F1 and 50 or above for F2, the nano-vectors were retarded completely (remaining in the well and had no movement within the gel) (Figure 4).

### 3.6. MiR-375 Conjugation Efficiency

The conjugation efficiency of F1/miR-375 and F2/miR-375 nano-plexes was evaluated using a Nanodrop spectrophotometer (Table 4). At a weight ratio of 250 or above for F1 and 50 or above for F2, the percentage of conjugation efficiency reached 100%. This result suggested that miR-375 was fully incorporated into both F1 and F2 due to their excellent loading efficiencies.

### 3.7. Lyophilization Study

Table 5 shows the effect of different cryoprotectants on the PS and PDI of lyophilized NSLCs/miR-375. Compared to the freshly prepared colloids, the lyophilized NSLCs/miR-375 had much larger PS. Meanwhile, Avicel increased the freeze-drying time and aggregation of nano-plexes due to inter-and intra-molecular hydrogen bonding, resulting in PS of 2320.31 ± 1103.20 nm for F1/miR-375 and 2656.22 ± 1066.04 nm for F2/miR-375, which were 10 and 17 times higher than the freshly prepared colloid, respectively. In comparison, the use of sucrose resulted in shorter freeze-drying process times and significantly smaller PS of 219.15 ± 9.85 nm in F1/miR-375 and 283.22 ± 7.06 nm in F2/miR-375 compared to the lyophilized samples with other cryoprotectants (*p* < 0.001). However, there was no significant difference in PS compared to the freshly prepared colloid in F1/miR-375 and F2/miR-375 (*p* > 0.05). Our finding is consistent with what was previously reported about sucrose’s ability to reduce the diffusion coefficient of water molecules and the rate of ice crystal growth [83]. Also, its stable amorphous form protects the NPs in a “pseudo-hydrated” form via hydrogen bonding, which provides a shield from ice crystals that may damage the NPs during lyophilization and later during re-dispersion [84]. The processing time of trehalose was longer than that of sucrose because trehalose does not have internal hydrogen bonding and needs to absorb water molecules from the surrounding environment due to its hygroscopicity, while Avicel possesses an affinity for water from the two hydroxyl dominant sides in its crystalline structure, and the other two sides of crystals are wetted by lipid NPs [85,86,87]. The PS and PDI of the reconstituted lyophilized F1/miR-375 and F2/miR-375 were then compared to samples frozen at −20 °C. The results showed no significant difference in PS between the frozen samples and the freshly prepared F1/miR-375 and F2/miR-375 colloid (*p* < 0.05). Collectively, our findings suggested that freezing miR nano-plexes in a regular freezer at −20 °C provides an alternative storage option that is less stressful than lyophilization, which can be complex and expensive.

### 3.8. Stability Studies

#### 3.8.1. Storage Stability Study

The storage stability of lyophilized and frozen at −20 °C nano-plexes was also investigated by changes in PS, ZP, and PDI for 3 months as shown in Table 6. The results showed that PS did not change significantly in both lyophilized and frozen formulations over the 3 months for F1/miR-375 and F2/miR-375 (*p* > 0.05). However, the lyophilized samples lost their positive surface charge, while the frozen samples were able to maintain their positive charges and miR-375 leakage was not observed throughout 3 months. The loss of positive surface charge in lyophilized nano-plexes might be attributed to the destabilization of the CS polymer chains, which occurred during the lyophilization process. This loss of positive charge and more neutral surface charges around lyophilized nano-plexes during storage might lead to aggregation due to the absence of an inter-particle repulsive force [88,89]. On the other hand, the frozen nano-plexes might have been able to maintain their positive charges due to the preservation of the polymer chains in their native conformation. Additionally, a narrowing of the PDI was observed after 3 months, indicating the presence of more uniform larger NPs. Overall, the nano-plexes remained stable with slightly larger PS and reduced ZP after 3 months and they exhibited good ability to reduce miR-375 expulsion during this storage period. Based on these findings, it is recommended to store the NLSC/miR375- nano-plexes in a freezer at −20 °C without lyophilization due to several advantages, including no applied external stress on the NPs under freeze-drying process, the availability and easy accessibility of refrigerators in research facilities.

#### 3.8.2. Serum Stability Study

To investigate the stability of the nano-plexes in a complex physiological environment and assess the ability of the studied nano-carrier to protect miR-375, F1/miR-375 and F2/miR-375 were incubated with 50% FBS for short (24 h) and long intervals (1 week) at 37 °C. The samples were then analyzed by agarose gel electrophoresis, as shown in Figure 5A,B. Naked miR-375, which showed a faint band, was severely degraded by RNase present in the FBS after 1 week, as indicated by the disappearance of the band in lane 1 of Figure 5B. In contrast, clearer bands with brightness were observed in lanes 2 and 3, representing F1/miR375 and F2/miR375, at both 24 h and 1 week. There was clear gel retardation of the F1 and F2 nano-vectors compared to free miR-375, indicating that the two CS NSLCs were maintained intact during the incubation period and effectively protected the miR-375 from degradation by serum endonucleases. This finding will be further confirmed through qualitative analysis using fluorescence microscopy and quantitative analysis using flow cytometry.

### 3.9. Transfection Efficiency

#### 3.9.1. Cellular Uptake and Intracellular Localization

The cellular uptake and intracellular localization of HepG2 cells was examined using fluorescence microscopy, to investigate the effectiveness of naked miR-375, F1/miR-375, and F2/miR-375 in delivering miR-375 into the cells (Figure 6A). The results showed that miR-375 in both F1 and F2 demonstrated a higher degree of transfection efficiency and re-expression of miR-375 compared to cells treated with naked miR-375. This was indicated by the greater intensity of the green fluorescent protein (GFP) marker in cells treated with F1/miR-375 and F2/miR-375. More interestingly, the results of both fluorescence microscopy (Figure 6B) and flow cytometry (Figure 7) together confirmed no significant decrease in the uptake of miR-375 after 24 h of contact with serum endonucleases. Meanwhile, both F1/miR-375 and F2/miR-375 even in the presence of 50% FBS pre-treatment could also be successfully absorbed by cells and displayed a stronger MFI than free miR-375.

#### 3.9.2. Quantitation of Cellular Uptake

Likewise, fluorescence microscopy data and the flow cytometry result (Figure 7) showed that cells transfected with F2/miR-375 nano-plex significantly exhibited the highest mean fluorescence intensity (MFI) than cells treated with free miR-375 (* *p* < 0.001). Meanwhile, groups treated with F1/miR-375 and F2/miR-375 had 1.9–2.8 times higher MFI than naked miR-375. These results confirmed that both F1/miR-375 and F2/miR-375 were far superior to the Turbofect transfection reagent in delivering miR-375. We concluded that these nano-formulations could be a promising tool for delivering miR-375 and enhancing its endogenous re-expression while maintaining its elevated concentration to exert its pharmacological action with sustained release and stability in treated cells.

### 3.10. In Vitro Anticancer Efficacy

The cytotoxicity rate of HepG2 cells after 48 hrs. of transfection was investigated based on MTT assay. Both F1 and F2 nano-formulations significantly exhibited low cytotoxicity, with 2% to 13.7% for F1 and 1.2% to 12.2% for F2 at a concentration of 10–500 μg/mL in HepG2 cells, and the IC50 was not reached in both cases (Figure 8). Thus, these data suggested that both CS-coated NSLCs can be applied as safe carriers for miR-375 nano-delivery. Additionally, the cytotoxicity was found to decrease with increasing CS concentration in the nano-formulation. Accordingly; F2 had a higher positive charge but less toxic effect on the tested cells, making it a more satisfactory formula in terms of safety, and was selected as the optimal formula.

In addition, as shown in Figure 8, the tumor suppression function of miR-375 in HCC has been demonstrated. For instance, miR-375 either free or nano-formulated via F1 or F2 significantly impaired HepG2 cell growth by 36.4%, 39.8% and 52.5%, respectively (** *p* < 0.001). Therefore, the more acidic formula F2 delivered miR-375 could enhance the highest cellular uptake and exhibit more cytotoxicity in transfected cells compared to miR-375 alone or DOX. More importantly, it significantly exceeded the IC50 with 1.4-fold higher cytotoxicity compared with free miR-375. These results collectively confirmed that miR-375 delivered by both F1 and F2 in treated cells was significantly showed excellent uptake efficiency and showed a drastic increase in miR-375 and inhibition of its down-stream oncogenic pathways and involved targets.

## 4. Discussion

HCC is the second most lethal tumor globally and most, if not all, of the currently available therapeutics still do not meet the intended outcomes and have restrictive toxicities [1,2,3]. Thus, the eradication and cure of this aggressive tumor necessitates the development of novel therapies and efficient delivery systems.

Tumor suppressor miRs have been intensively reported to hold promise as powerful therapeutic agents for a broad variety of tumors including HCC [10,12,15,18,90,91,92]. This is due to their ability to specifically silence the expression of cancer-related genes and pathways that underlie tumor formation and progression. Accordingly, miRs exert their regulatory effects by nearly perfect base pairing with target genes leading to mRNA cleavage or translational repression [5,7]. 

Growing evidence has indicated that miR-375 is one of the most down-expressed miRs and its downregulation crucially switches on the development and progression of many malignant tumors including HCC [11,12,13,21,22,23]. While its overexpression has been recognized to predominantly suppress all the core hallmarks of HCC [24,27,28,29]. The involvement of miR-375 in each of these hallmarks proceeds by directly targeting several important oncogenes-driven hepatocarcinogenesis like *AEG-1*, *MDR1*, *P-gp*, *YAP1*, *ATG7*, *ATG14*, *Bcl-2*, *SIRT5*, and *AKT/Ras* [14,22,25,30,93]. Therefore, miR-375 has been confirmed to simultaneously play a strong anti-HCC effect.

The clinical translation of miRNA therapy mainly requires the development of a specific delivery system since free miRNA cannot enter tumor cells efficiently, and may be vulnerable to nuclease degradation and poor endosomal release [31,32]. In this regard, a variety of viral and non-viral carriers have been reported for miRNA delivery in cancer cells [33,34]. However, nano-carriers have been largely perceived increasing development over their viral counterparts due to their huge advantages as functional vehicles [18,35,36,37,38,39]. Among this plethora of nano-carriers, LBNPs have shown great improvement in the treatment outcomes of anti-neoplastic agents [94,95]. This backs to LBNPs large-scale production, superior biocompatibility, lower immunogenicity, and non-toxicity due to their biodegradation, high drug loading, tumor targeting, and rapid cellular uptake [96,97]. Interestingly, LBNPs have demonstrated potential capabilities to encapsulate, deliver, and stabilize therapeutic miRs against enzymatic degradation and prolong their circulation half-life time as well [40]. Thus, miRNA-based therapies delivered with LBNPs, namely nano-miRNAs, represent promising anti-tumor strategies [15,16,17,18,19,98]. 

NSLCs have been extensively shown to possess various advantages such as enhanced physical stability over time, higher drug loading capacity, negligible drug leakage, controlled release due to the diffusional barrier of the solid lipid nanostructure, and long metabolic cycles [99]. Furthermore, the bio-distribution of NSLCs has been easily modulated by different surface modifications. Thereby, they could achieve site-specific targeting of the tumors for better efficacy and reduced dose-related toxicity [43,54,100,101,102]. 

Various nano-carriers have been developed for enhancing HCC control by therapeutic miR-375. Examples included fabrication of cationic lipid-coated cisplatin/miR-375 NPs [103], lipid-coated hollow mesoporous silica and doxorubicin (DOX)/miR-375 NPs [104], DOX and miR-375 co-loaded into lipid-coated calcium carbonate NPs [105], sorafenib (SOR) and miR-375 were co-loaded into lipid-coated calcium carbonate NPs [106], DOX and miR-375 were co-delivered by liposomes [107]. All these miR-375 NPs acted as potential therapeutic agents for HCC cells by the significant increase in tumor uptake and inhibiting all malignant characteristics. Nevertheless, these NPs were found to have certain restrictive limitations like short-term effects, immunogenicity, and toxicity [12]. Also, the specific regulatory mechanism of miR-375 in the presence of these NPs is still unsatisfactory [108]. Therefore, there is an urgent need to continuously optimize the delivery vehicles and dose of this miR for maintenance of its therapeutic properties.

Accordingly, the importance of miR-375 replacement therapy for HCC attracted our focus to the further development of highly potent and scalable delivery strategies that help miR-375 to elicit stronger anti-tumor effects. Our study is the first to successfully fabricate two cationic NSLCs for transfer of miR-375 vector into HepG2 cells and proved its therapeutic efficiency. Initially, we optimized its nanostructures which are composed of solid/liquid lipids and CS coats on surfaces to enhance the biocompatibility and cationic properties which ensure the electrostatic linkage with the loaded miR-375 and importantly safeguard it against degradation.

It has been reported that HLB value, melting point, and solubility of lipids are the main parameters of its selection for NSLC preparation [100]. Precirol ATO-5 and oleic acid were the solid and liquid lipids that formulated our NSLC at a ratio of 1:1 and dramatically had the smallest PS, strongest ZP, and modest PDI. Whereas when the content of either lipid was exceeded, the PS increased abruptly (Table 2). Our explanation for the PS increase is in agreement with the previous report showing the expulsion of the excess lipid [109]. Considerably, it has been reported that the HLB value and optimized amount of the surfactant relative to the lipid in NSLC preparation are necessary for its selection. Meanwhile, its HLB value should be equal to or greater than the required HLB value of lipid such that its value should be more than 10 [110,111]. In addition, it is commonly used by only 1–5% to stabilize the dispersion of NSLC by minimizing its aggression and importantly preventing the cell death effect [109]. In line with this, for our two NSLCs, we used 3% surfactants; Gelucire 50/13 and Kolliphore^®^ with HLB values 11 and 29. Interestingly, both surfactants could provide optimized enhancement on the surface of each NSLC which allowed its binding to the nano-lipid chains and ultimately there was an overall reduction in PS, an increase in surface area, and an improvement of the biocompatibility via stable ZP and PI. 

Chitosan (CS), which is a natural and cationic polysaccharide, has been utilized by 0.5% and 1.5% to coat the surface of our NSLCs and gain benefit from its reported biocompatibility, biostability, biodegradability, target specificity [62,66,69,112,113,114,115,116,117]. Moreover, as we presented its promoted ability to form electrostatic nano-complex with miR-375 in the presence of optimized acidic pH enhance its intracellular stability as well. 

Various factors have been studied by optimizing the amount of lipids and CS in our prepared NSLCs to improve their properties [118,119,120]. Although all nano-formulations had reasonable properties (Table 2), F1 and F2 were specially chosen for this study because they displayed the best physio-chemical properties, stability, conjugation efficiency, and importantly the desired cell death activity by miR-375 in target HepG2 cells.

According to Table 3, the two CS-coated NSLCs, F1 and F2, had PS 65.2 nm and 100.2 nm with a ZP of 32.5 mV and 61.8 mV, respectively, and accepted PDI which indicated no agglomeration of both cationic nano-formulations due to the increased repulsive force between particles and their spatial blocking as well. Besides, both F1 and F2 were found to exhibit spherical structures as shown in Figure 2 of TEM analysis. Also, it indicated the nanostructure emulsion of lipid and aqueous compartments which enabled the easy loading of miR-375.

The overall physicochemical data suggested that both F1 and F2 could nano-formulate the anionic miR-375 and stabilize its conjugation efficiency as well via electrostatic interaction. For instance, The PS, ZP, and PDI of F1/miR-375 were 207.3, 7.7, 0.37 and that of F2/miR-375 was 243.17, 5.8, and 0.31. In addition, these characterizations were confirmed by TEM and FTIR. The results of the electrophoretic mobility indicated that miR-375 was retarded with an increasing amount of CS 0.5% in F1 and 1.5% in F2, and both remained at the top of the gel at a weight ratio of 250 and 50, suggesting that F1 and F2 formed nano-plexes completely with miR-375 at those ranges (Figure 4). In addition, the importance of surfactants in F1 and F2 nano-formulations was in agreement with the Barzegari et al. (2019) study, which showed that PEG blocks can partially affect the lack of motion of the miR nano-plexes along the gel [121]. In addition, the formed nano-plex was more efficient in F2 with a higher percentage of CS and this may be due to the formation of more entanglements between CS and the double-stranded miR-375 vector. Furthermore, the significantly high CE, which reached 100% (Table 4) at the mentioned weight ratios, suggested that both F1 and F2 are highly secure nanocarriers to simultaneously nano-formulate the cargo and thus prevent its possible leakage. This finding was confirmed by the stability test of F1/miR-375 and F2/miR-375 nano-plexes which could attain longer storage stability up to 3 months (Table 6). Remarkably, these storage stability results indicated the potential ability of our two CS-coated NSLCs to overcome the main difficulty of SLNs in terms of the polymorphic modification that occurs on its crystal structure during storage time [51,53]. Correspondingly, our NSLCs could reduce polymorphic modifications, and subsequently increase miR-375 loading capacity and stability during 3 months of storage. Also, they could stabilize PS, and shape, and prevent particle aggregation. 

It has been shown that the colloidal stability of NPs in complex physiological media is very challenging and the miRNA is labile in serum [32,122]. Accordingly, the representative protective effect of our fabricated CS-coated NSLCs on miR-375 plasmid from endonuclease degradation and maintaining its stability were examined by subjecting F1/miR-375 and F2/miR-375 to 50% FBS as a model enzyme. Then, Gel electrophoresis confirmed that these nano-plexes are stable and well tolerated even after 1 week (Figure 5). Additionally, there was observed fluorescence in treated HepG2 cells with these nano-plexes which consequently verified the integrity of miR-375 with time (Figure 6). 

The importance of the ingredients used in our NSLCS has been noted in previous reports as we discussed here. Precirol is a surface-active partial glyceride that facilitates emulsification and the formation of a solid matrix with a perfect lattice [123]. In addition, its reported formulations showed large PS due to the large spaces among its particles and increasing viscosity. Moreover, its formulations exhibited a negative charge due to the liberation of anionic fatty acids [124]. Various SLNs consisting of Precirol had been developed to form lipoplexes with DNA plasmids and chemotherapeutics for their delivery and mediating cytotoxic effects in cancer cells. However, they were found to be insufficient for drug loading or exhibited drug expulsion [125,126,127,128].

Up on Precirol incorporation with liquid lipid, massive crystal-order disturbances have been created sufficient to accommodate drug molecules. This desirable change is consistent with our NSLCs which had smaller PS and moderate polarity, resulting in increasing the miR-375-loading capacity (Table 3). In addition, these NSLCs are characterized by high viscosity and strong interfacial film due to the presence of Precirol and thereby they could control the release capability of miR-375 to sustain its long-term effect in target cells. Thus, our study is consistent with that of Chen et al.’s (2010) study which showed the success of NSLC containing Precirol in increasing lovastatin loading efficiency, delivery, and therapeutic effects [123].

Oleic acid (OA), an unsaturated fatty acid, has been reported to be included in the content of various NSLCs. These nanocarriers have been shown to exhibit a dramatic decrease in PS by increasing the OA amount and they significantly enhanced the delivery and transfection efficacy for both siRNA and miRNA [129]. Meanwhile, this study by Wang et al. (2013) fabricated LBNP containing OA which efficiently delivered miR-122 in HCC cells. Another NSLC consisting of OA had successfully co-delivered DOX and SOR and it induced immunogenic cell death and tumor microenvironment (TME) remodeling [129]. Additional galactosylated NSLC containing OA had been demonstrated as a targeted delivery of 5-fluorouracil (5-FU) in HepG2 cells. This NSLC had PS 139.2 nm, ZP–18 mV, loading efficiency of 34.2% and importantly it caused cytotoxic effects by reducing the 5-FU dose to half its concentration [109]. In comparison with all these reports of effective NSLCs, our cationic NSLCs achieved better CE, long-term physical stability, transfection efficiency, sustained stability, and stronger anti-cancer activities of F1/miR-375 and F2/miR-375 nano-plexes in target HepG2 cells. All these effects mainly refer back to the nanostructures of F1 and F2. 

Lyophilization has the advantage of enhanced stability of pharmaceutical nano-products in a freeze-and-dry state [130]. However, it often requires the addition of cryoprotectants to prevent NP aggregation [79,80]. In line with this, our study showed that the lyophilized F1/miR-375 and F2/miR-375 in the presence of sucrose exhibit the smallest PS of 283.2 ± 7.063 nm and 219.1 ± 9.854 nm, modest ZP, and PDI, respectively compared to lyophilized samples in presence of other cryoprotectants. Thus, sucrose serves as the best cryoprotectant, and its lyophilized samples were very similar to the non-freeze-dried colloid. Unexpectedly, we also presented for the first time that the NSLC/miR-375 nano-plexes could be more stable and easier to store by freezing at −20 °C based on the obtained physicochemical properties which are much closer to the fresh colloid (Table 6).

Regarding cellular uptake, our finding is consistent with all transfection efficiency validation [104,131]. Meanwhile, the cationic F1 and F2 nano-formulations could deliver the therapeutic payload to target cells in a controlled and stabilized manner. Accordingly, the green fluorescence which is predominately distributed in the cytoplasmic region of F1/miR-375 and F2/miR-375 treated cells compared with free miR-375 indeed referred to the enhanced internalization of miR-375, its efficient release and upregulated expression of its mature form (Figure 6A).

Moreover, there was a significant increase in MFI in cells treated with F1/miR-375 and F2/miR-375 nano-plexes compared to those treated with free miR-375 (* *p* < 0.001) (Figure 7). More importantly, both nano-plexes could protect miR-375 against degradation and maintain its nano-delivery and re-expression in treated cells. Meanwhile, following 24 h incubation with 50% FBS and cell transfection, there was observed fluorescence and no significant change in MFI in the treated cells compared to those treated with free miR-375.

Safety is a critical issue in the development of miR therapeutics and cationic LBNP delivery has been debated due to its probable cytotoxicity [132,133]. However, our prepared blank; F1 and F2 significantly exhibited negligible cytotoxicity in HepG2 treated cells even at a high concentration of 500 µg/mL, in comparison with negative control (Figure 8). Interestingly, at this high concentration, the determined % of cytotoxicity of F1 and F2 (13.7% and 12.2%) were significantly lower than that of chitosan-lipid nanocarrier (nearly 15%) in a variety of tumor cells [134]. Our obtained results back to the strategic selection of NSLC components and its optimal combination with M.Wt CS which alone was shown to need a very high concentration of 1850 μg/mL to reach IC50 in HepG2 cells [82]. Overall, our developed cationic nano-formulations are unlike the previously reported ones [135,136] by exhibiting diminished cytotoxicity and biodegradation fate in target cells.

Since proliferation is the most distinctive and indispensable hallmark of cancer cells to maintain their viability [137,138], therefore, we were interested in evaluating the anti-proliferative effects of our two miR-375 nano-plexes in HepG2 cells. Excitingly, we showed for the first time the exact percentage of cell viability inhibition by F1/miR-375 and F2/miR-375 nano-plexes compared with free miR-375, respectively (** *p* < 0.001) (Figure 8). In addition, this free miR-375 was found to significantly suppress cell proliferation better than previously reported [26]. Considerably, the enhanced effect of our therapeutic nano-plexes has been confirmed by comparing it with 11 µg of DOX, the previously reported IC50 in HepG2 cells [81]. We found more potent anti-proliferative activity in the F2/miR-375 treated cells although its IC50 value (0.2 µg) is 55 times lower than that of DOX. This means F2 could enhance much stronger and more durable inhibitory effects of miR-375 in the presence of higher CS concentration. Collectively, our data support the previous studies that genomic loss of miR-375 promotes advanced proliferation in cancerous cells [21,139], whereas its overexpression by NPs can exert predominant improvement of the anti-HCC effects [40]. Besides, the overall findings confirmed our main objective that the miR-375 either free or nano-formulated is taken up by HepG2 cell and is stably re-expressed from the generated pEGP-pre-miR-375 vector. In turn, it is efficiently released in the cytoplasm to function properly in the RNAi machinery and continuously mediates its tumor suppression via the reduction of its downstream oncogenic targets and networks of HCC.

Focusing on the mechanism of miR-375 delivery, the free miR is well-known to be taken in target cells by diffusion, however, the immediate entry of its large volumes may prevent further import [121]. On the contrary, the nano-formulated miR has been showed to penetrate cells by endocytosis [104,132]. Likewise, our miR nano-plexes, F1/miR-375 and F2/miR-375, caused uniformity in miR-375 absorption into HepG2 cells and its slow release for the long term. In addition, this enhanced internalization of miR-375 is in agreement with previously reported LBNPs due to the cell membrane affinity to nano-lipids of both F1 and F2 facilitating endocytosis, escape from endosome/lysosome thus avoiding degradation and enhancing enormous re-expression of functional miR-375 [104]. Moreover, both F1 and F2 like previously reported NSLCs specifically transport the drug payload to the tumor cells through passive targeting, active targeting, and co-delivery mechanisms [51,124]. The passive mechanism called the EPR effect takes advantage of the tumor microenvironment [140]. Thus, NSLCs have shown high permeability to traverse across leaky vasculature and passively accumulate in tumor cells [53]. In this context, NSLCs have been widely used for effective targeted cancer chemotherapeutics [141,142,143]. Our NSLCs are particularly used for effective miR-375 delivery owing to the acidic pH of both F1 and F2 which are selective to tumor cells. The active mechanism of NSLCs has been shown to involve its surface functionalization with different ligands or miR [144]. A variety of NSLCs have been developed to improve loading efficiency, targeting capability, and effective gene silencing of multiple miRs through endocytosis. Examples of these miRs included miRNA-125-a-5p, anti-miR-221, miR-34a, let-7-a, and anti-miR-21 [57,58,59,60,61]. Our suggested co-delivery mechanisms of miR-375 by the fabricated NSLCs to target HepG2 cells are also aided by the CS coat. This is because our result was in line with previous studies which have suggested that cationic CS NPs possess penetration enhancer properties that allow them to have an electrostatic affinity for negatively charged cell membranes, leading to their selective accumulation in tumors, such as liver tumors by passive and active targeting [53,66,68,69,112,113,114,115,116,145]. Overall, as shown from our collected data, the two fabricated CS-coated NSLCs pave promising evidence to make them ideal strategies for miR-375 replacement therapy in HCC cells. 

## 5. Conclusions

In summary, our study emphasizes the significance of the two fabricated CS-coated NSLCs/miR-375 as smart therapeutic strategies for HCC via the innovative and optimized NSLCs and the stable miR-375 expression construct. These cationic NSLCs exhibited desirable physicochemical properties corroborating their high compatibility and conjugation efficiency for the miR-375 vector which reached 100%, storage stability without significant leakage of miR for up to 3 months, and serum stability against degradation after 1-week incubation. Furthermore, these nano-carriers were non-toxic in target HepG2 cells, enhancing its biosafety for miR-375 nano-delivery. Interestingly, the NSLCs improve the specific delivery of miR-375 to cells through passive and active targeting and also enhance the intracellular stability of the released miRNA and its therapeutic effects. The miR-375 nano-plexes, especially F2/miR-375, exhibited excellent anti-proliferative efficacy and exerted predominant inhibition of tumor cell growth higher than the free form of miR-375 and the standard treatment using doxorubicin. Therefore, our novel nano-delivery systems are highly efficient to provide promising breakthroughs for further in vitro and in vivo applications shortly.

## 6. Patents

The authors declare that there is a submitted Egyptian patent with ID EG/P/2023/708.

## Figures and Tables

**Figure 1 pharmaceutics-16-00494-f001:**
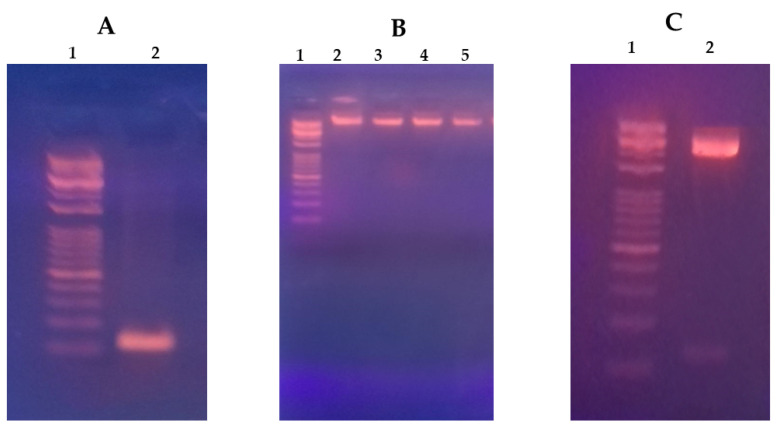
MiR-375 vector expression construct. (**A**) Lane 1 represents the DNA marker (5000–100 b.p), and lane 2 represents the PCR purified pre-miR-375 sequence (127 b.p). (**B**) Lane 1 represents the DNA marker, lanes 2–5 represent samples of purified pEGP-miR-375 plasmid of approximate size 5000 b.p following ligation of miR-375 sequence, transformation in DH5 alfa bacterial strain and finally plasmid maxiprep purification. (**C**) Lane 1 represents the DNA marker, and lane 2 represents double digested pEGP-miR-375 plasmid showing digested vector (upper band) and digested miR-375 sequence (lower band) to confirm the overall expression construct.

**Figure 2 pharmaceutics-16-00494-f002:**
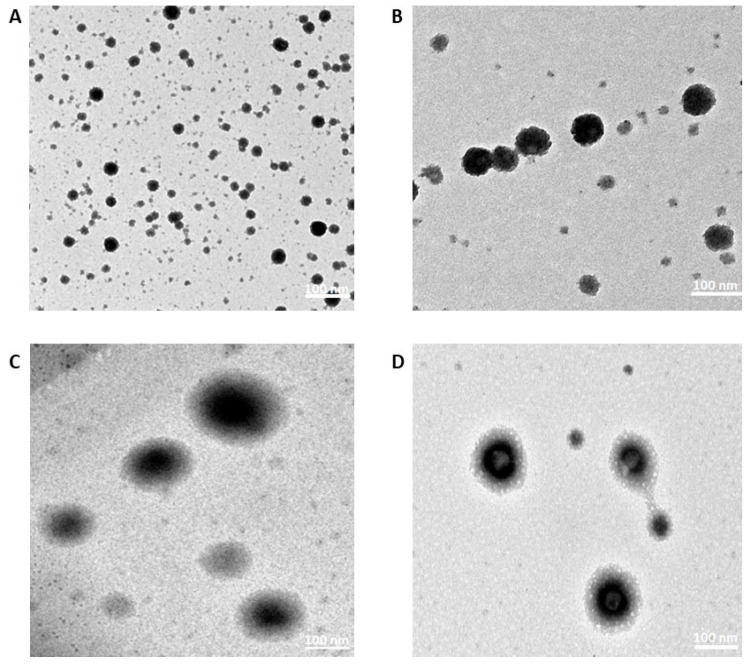
TEM images of (**A**) blank F1, (**B**) blank F2, (**C**) F1/miR-375 weight ratio 250:1, and (**D**) F2/miR-375 weight ratio 50:1. Magnification is 100,000×.

**Figure 3 pharmaceutics-16-00494-f003:**
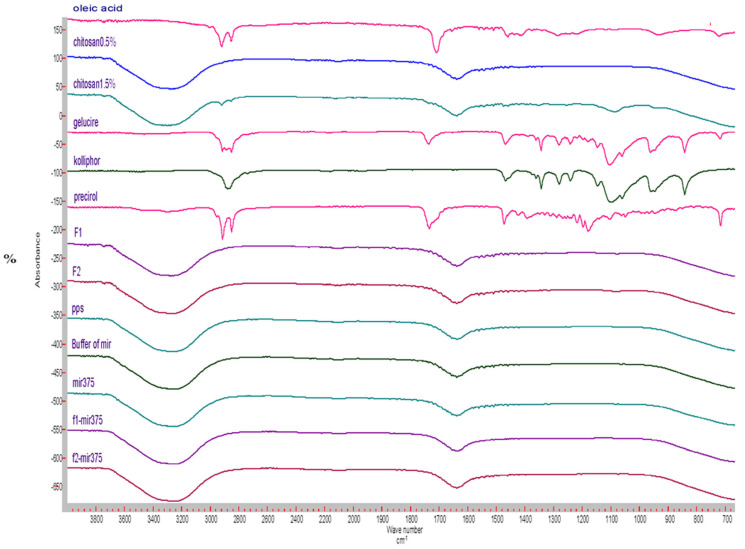
FTIR graph of the individual components of the two nano-formulations, blank F1, F2, and conjugated miR-375.

**Figure 4 pharmaceutics-16-00494-f004:**
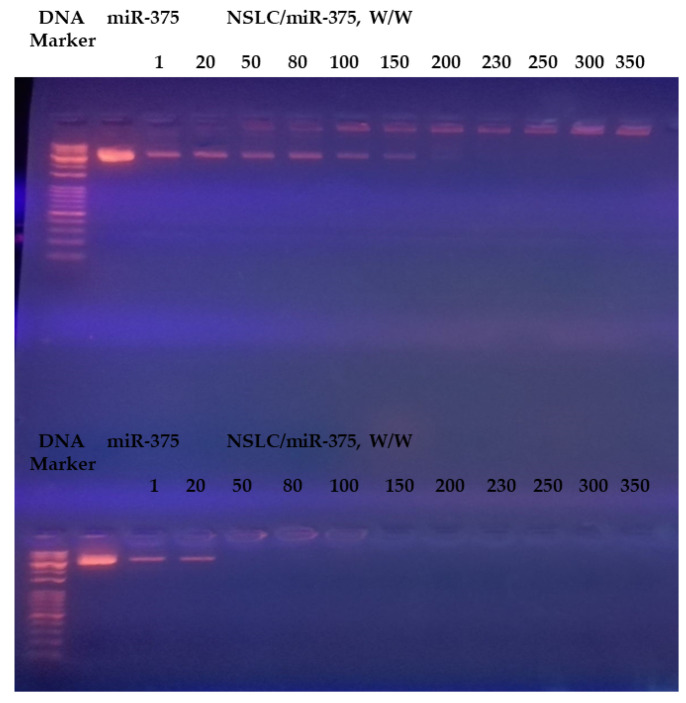
Gel retardation analyses of (**top**) F1/miR-375, and (**bottom**) F2/miR-375 were prepared at different weight ratios to determine the maximum effective concentration of CS.

**Figure 5 pharmaceutics-16-00494-f005:**
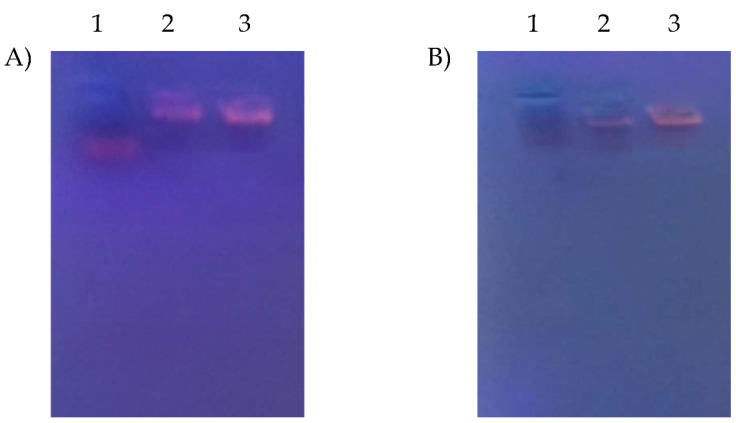
Analysis of serum stability using gel retardation. (**A**) After 24 h. of incubation in 50% FBS. Both nano-plexes showed no effect of serum, respectively. (**B**) After 1 week of incubation with 50% FBS. Lane 1 represents 1 µg of naked miR-375, lane 2 represents F1/miR-375 with a weight ratio of 250:1, and lane 3 represents F2/miR-375 with a weight ratio of 50:1. There is a verified serum stability of both nano-vectors and there is a clearer gel retardation of F2 nano-vector which confirms its stability even in presence of serum.

**Figure 6 pharmaceutics-16-00494-f006:**
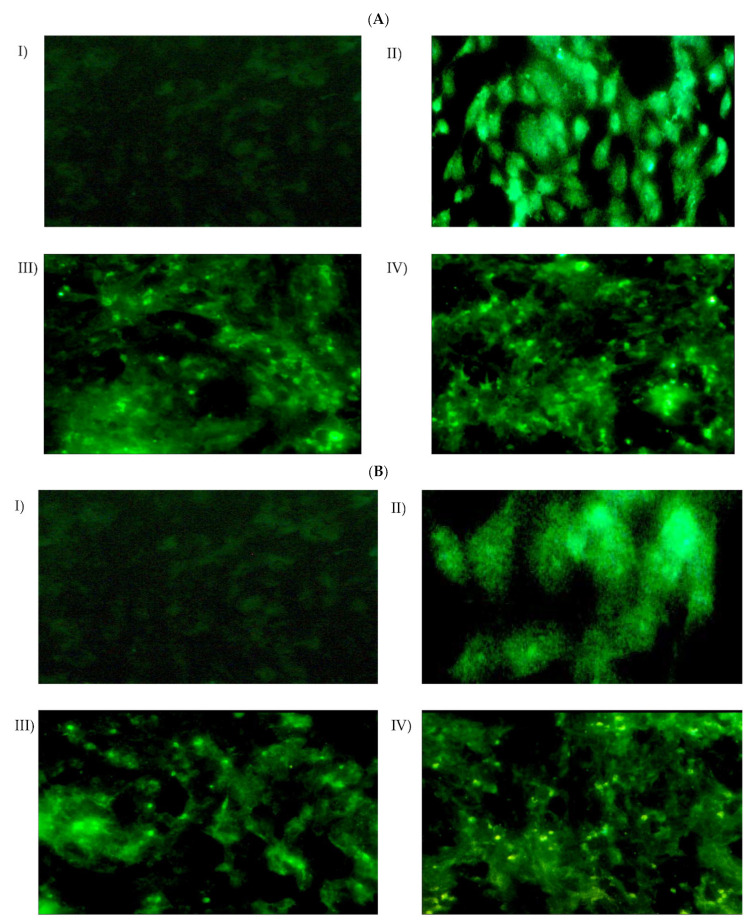
(**A**) Fluorescence micrographs of CS-coated NSLCs with miR-375 in HepG2 cells. (**B**) Following incubation in 50% FBS for 24 h and treatment in HepG2 for 48 h at 37 °C. (**I**) Control cells without treatment, (**II**) Transfected cells by naked miR-375 vector, miR-375 re-expression in cells was revealed by the co-expressed GFP, (**III**) Transfected cells by F1/miR-375 at ratio 250/1, and (**IV**) Transfected cells by F2/miR-375 at ratio 50/1. All of the naked miR-375, F1/miR-375, and F2/miR-375 were localized in the cytoplasm (Magnification: 60×). The scale bar is 50 µm in all images.

**Figure 7 pharmaceutics-16-00494-f007:**
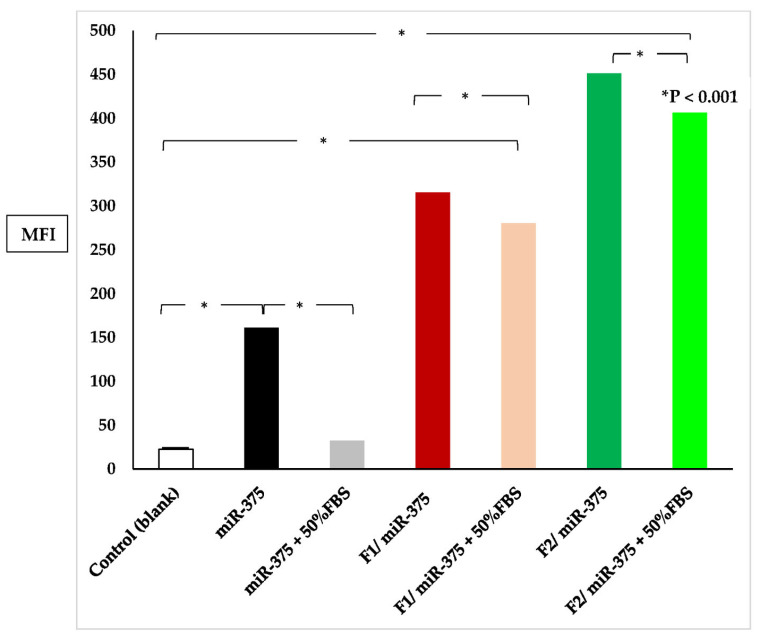
The mean fluorescence intensity (MFI) was measured in the presence and absence of 50% FBS. The measurements were taken for miR-375 (positive control), F1/miR-375 (at a ratio of 250/1), and F2/miR-375 (at a ratio of 50/1). The measurements were also taken for miR-375-FBS, F1/miR-375-FBS, and F2/miR-375-FBS to confirm the serum stability of the two nano-plexes compared to naked miR-375.

**Figure 8 pharmaceutics-16-00494-f008:**
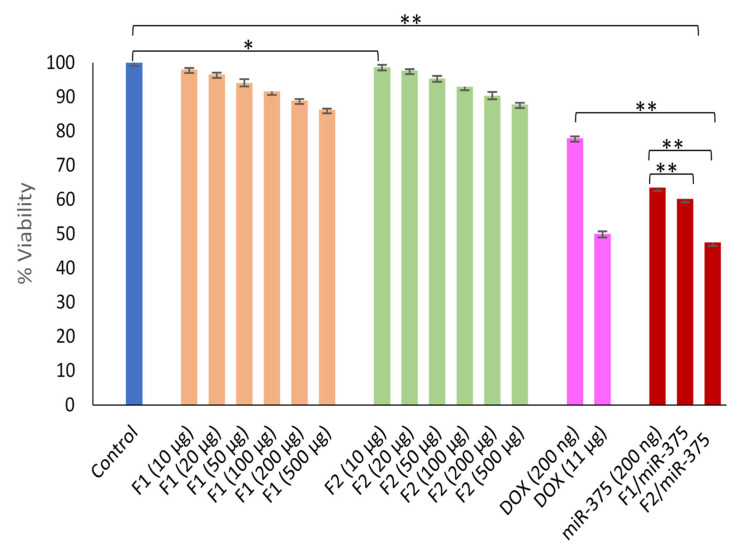
Investigation of non-cytotoxicity impact of empty F1 and F2 formulations compared to negative control. While the anti-cancer activity of nano-plexes F1/miR-375, and F2/miR-375. Naked miR-375 and Doxorubicin were also included as positive controls—treatment after 48 hrs. in HepG2 cells. * *p* > 0.05; ** *p* < 0.001.

**Table 1 pharmaceutics-16-00494-t001:** Nano-formulation factors for optimization CS-coated NSLC.

	Lipid (mg)		Surfctant (mg)		Water (mL)	Chitosan (%)	pH
Studied Factors	Solid	Liquid					
	Precirol ATO-5	Oleic Acid	Gelucire 50/13	Kolliphore P188			
Effect of solid: liquid lipid ratio	100	300	70	130	10	1.0	4.0
	134	266					
	200	200					
	266	134					
	300	100					
	320	80					
Effect of surfactant concentration	275	275	17	33	10	1.0	4.0
	247	247	35	71			
	194	194	71	129			
	141	141	106	212			
Effect of chitosan concentration	141	141	106	212	10	0.5	4.0
						1.0	
						1.5	
						2.0	
Effect of pH of chitosan solution	141	141	106	212	10	1.0	4.0
							4.5
							5.0
							5.5

**Table 2 pharmaceutics-16-00494-t002:** Factors influence particle size (PS), zeta potential (ZP), and polydispersity index (PDI) of the of the CS coated NSLCs (*n* = 3).

Factors	Ratio (Solid: Liquid Lipid)	Formulation Code	PS Z-Average (nm)	ZP (mV)	PDI
Effect of Precirol: Oelic acid ratio	1:3	A	243.22 ± 10.86	50.46 ± 3.33	0.48
	1:2	B	157.81 ± 5.16	38.72 ± 6.80	0.45
	1:1	C	116.72 ± 3.10	43.25 ± 4.71	0.42
	2:1	D	199.43 ± 8.14	31.33 ± 2.40	0.17
	3:1	E	246.06 ± 11.72	52.17 ± 5.34	0.55
	4:1	F	137.84± 8.03	59.41 ± 6.61	0.43
Effect of Gelucire/Kalliphore concentration (*w*/*w*)	0.5%	G	138.92 ± 9.67	46.72 ± 3.33	0.35
	1.0%	H	130.53 ± 5.02	46.56 ± 4.80	0.46
	2.0%	I	116.76 ± 3.10	43.28 ± 4.71	0.42
	3.0%	J	101.03 ± 6.14	43.71 ± 3.40	0.23
Effect of chitosan concentration (*w*/*w*)	0.5%	K	89.45 ± 9.67	45.91 ± 4.03	0.31
	1.0%	L	101.02 ± 6.14	43.75 ± 3.40	0.23
	1.5%	M	100.27 ± 3.10	61.83 ± 2.74	0.40
	2.0%	N	103.23 ± 5.24	48.34 ± 5.01	0.41
Effect of Chitosan solution pH	4.0	O	98.17 ± 8.65	28.03 ± 2.33	0.43
	4.5	P	65.24 ± 6.02	32.54 ± 3.86	0.25
	5.0	Q	68.22 ± 2.13	16.57 ± 3.04	0.22
	5.5	R	77.62 ± 6.01	9.28 ± 4.12	0.19

**Table 3 pharmaceutics-16-00494-t003:** Physico-chemical characterization of the two nano-plexes bearing miR-375 (*n* = 3).

Characterization	PS (nm)	ZP (mV)	PDI
F1 (blank)	65.2 ± 6.02	32.5 ± 3.86	0.25
F1/miR375 *	207.33 ± 8.08	7.7 ± 1.02	0.37
F2 (blank)	100.2 ± 3.10	61.8 ± 2.74	0.40
F2/miR375 **	243.12 ± 9.33	5.8 ± 1.34	0.31

* F1/miR375 weight ratio 250:1. ** F2/miR375 weight ratio 50:1. The results are presented as means of triplicate reading of each CS-coated NSLC and presented as mean ± standard deviations (*n* = 3).

**Table 4 pharmaceutics-16-00494-t004:** Conjugation efficiency of the nano-plexes based on Nanodrop assessment (*n* = 3).

F1/miR-375	Conjugation (%)	F2/miR-375	Conjugation (%)
1:1	70.39	1:1	92.50
20:1	75.22	20:1	98.13
50:1	80.16	50:1	100
80:1	85.57	80:1	100
100:1	90.60	100:1	100
150:1	93.79	150:1	100
200:1	95.90	200:1	100
230:1	97.0	230:1	100
250:1	100	250:1	100
300:1	100	300:1	100
350:1	100	350:1	100

**Table 5 pharmaceutics-16-00494-t005:** The impact of several cryoprotectants on PS and PDI of lyophilized F1/miR-375 and F2/miR-375 nano-plexes compared to freshly prepared F1/miR-375 and F2/miR-375 colloids, as well as those frozen at −20 °C (*n* = 3).

Cryoprotectants	Freeze-Drying Time (h)	PS (nm)		PDI	
		F1/miR-375	F2/miR-375	F1/miR-375	F2/miR-375
Avicel	6	2320.31 ± 1103.20	2656.22 ± 1066.04	0.42	0.46
Mannitol	6	872.27 ± 103.14	629.25 ± 81.53	0.58	0.53
Lactose	6	445.72 ± 110.03	574.74 ± 104.81	0.44	0.57
Trehalose	5.5	303.31 ± 79.13	330.50 ± 103.71	0.17	0.49
Sucrose	3.5	219.15 ± 9.85	283.22 ± 7.06	0.40	0.41
Freeze at −20 °C	-	251.34 ± 3.38	264.65 ± 7.46	0.43	0.56
Freshly prepared colloid; CS-coated NSLCs/miR-375	-	207.33 ± 8.08	243.12 ± 9.33	0.37	0.31

**Table 6 pharmaceutics-16-00494-t006:** Three-month stability test of the lyophilized miR-375 nano-plexes (sucrose) (A), and frozen at −20 °C miR-375 nano-plexes (B), compared to the freshly prepared miR-375 nano-plexes colloids in terms of PS, ZP and PDI (*n* = 3).

Duration	PS (nm)				ZP (mV)				PDI			
	F1		F2		F1		F2		F1		F2	
	A	B	A	B	A	B	A	B	A	B	A	B
Fresh NSLC /miR375 Colloid	207.33 ± 8.08		243.12 ± 9.33		5.8 ± 1.34		7.7 ± 1.02		0.36		0.31	
Day 7	248.66 ± 6.35	281.25 ± 2.31	279.15 ± 8.95	289.47 ± 9.51	4.33 ± 6.01	7.36 ± 2.03	5.36 ± 3.02	5.34 ± 2.50	0.41	0.36	0.54	0.51
1 month	270.05 ± 5.56	269.36 ± 5.87	286.44 ± 4.36	300.89 ± 5.68	3.24 ± 3.02	5.22 ± 6.01	4.51 ± 4.11	4.97 ± 1.57	0.37	0.42	0.51	0.45
2 month	258.36 ± 7.26	274.32 ± 3.02	300.57 ± 3.65	302.27 ± 4.53	0.36 ± 2.03	4.03 ± 4.23	2.34 ± 1.98	5.03 ± 1.11	0.39	0.39	0.43	0.41
3 month	280.05 ± 8.32	285.61 ± 5.03	305.78 ± 2.98	310.39 ± 4.56	−4.56 ± 3.01	4.39 ± 0.42	−2.09 ± 1.22	3.85 ± 2.22	0.40	0.45	0.66	0.58

## Data Availability

The dataset supporting the conclusions of this article is available at request from the corresponding author.

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
