# Peer review of "Preparation and Optimization of MiR-375 Nano-Vector Using Two Novel Chitosan-Coated Nano-Structured Lipid Carriers as Gene Therapy for Hepatocellular Carcinoma"

_pharmaceutics, 2024, doi:10.3390/pharmaceutics16040494_

Round 1
Reviewer 1 Report (New Reviewer)
Comments and Suggestions for Authors
Dear author,
The paper entitled “Preparation and Optimization of MiR-375 Nano-vector using Two Novel Chitosan Coated/Nano Structured lipid Carriers as Gene Therapy for Hepatocellular carcinoma” has been intensively reviewed and evaluated. Although present study was considered an interesting study, there were some points that need to be revised. Hereby, I would like to present my suggestions and revisions.
Revision_1: abstract must be written following the journal's templates
Revision_2: it is clearly written in all its parts, but in my opinion, we need to amplify the introductory part regarding chitosan. I suggest this paper https://doi.org/10.3390/pharmaceutics14050942
Revision_3: table 5. why are the freeze-drying times different?
After these small modifications, the work can be published
Author Response
Here in attach the response letter.

Reviewer 2 Report (New Reviewer)
Comments and Suggestions for Authors
1. The authors did not keep the lipid concentration constant while studying the effect of SAA concentration. Why?
2.In line 180: Does twice mean 2-fold dilution?If yes, was this dilution enough? As LBNPs usually need higher dilution.
3.Why wasn't formulation L selected despite the insignificant difference in PS and the smaller PDI compared to formula M?
4. Why do you think the charge increased with 1.5% chitosan then decreased again with 2%?
5. Why did the charge decrease to that extent with this slight change in pH from 4.5 to 5?
6. In lines 370-371: The authors marked the superiority of F2. How was this judged, although there is an insignificant difference in zeta potential between F1 and F2 following coating? Also, please clarify the weight ratio in Table 3.
7. Why didn't the authors use the same weight ratio of the two formulations for a fair comparison?
8. Line 436: The sentence is incomplete.
9. Is the size obtained by trehalose significantly different from sucrose? It seems to be insignificant.
10. In the stability study, the authors recommended storage at -20 C but the supplementary figure shows a drastic change in zeta potential. How could this significant decrease in positive zeta potential be explained?
11. The first page of the discussion is a repetition for what has already been said in the introduction and needs to be reduced.
12.The authors discussed the safety of the formulations through investigating its cytotoxicity on HepG2 which is a cancer cell line. This is better done on normal cells to calculate selectivity index of the nanoparticles. Please justify.
13. Why isn't there a section named discussion?
14. Line 561: Please check sentence structure
15. In line 757 the authors mentioned that they investigated cell proliferation. How was cell proliferation tested?
16.Line 792:correct pH to start with a small p
17. With the positive charge of CS almost totally masked, do you think it is still in contact with the cells acting as a mucoadhesive as mentioned in line 800?
18. The supplementary figure is not very clear
19. On what basis was the miRNA concentration used in the cytotoxicity study selected?
Comments on the Quality of English Language
Sentence structure needs revision to make the text more clear.
Author Response
Here in attach the response letter and edited manuscript.

Reviewer 3 Report (New Reviewer)
Comments and Suggestions for Authors
Dear Authors,
The study is interesting and novel.
I have the following remarks:
The abstract needs edition– lines 25,31. The abstract does not need marked subsections.
The article contains following sections: Introduction; Materials and methods; Results; and Conclusions. The authors discussed the results on 4 pages but they haven’t marked section Discussion – should be corrected.
Introduction is well written and provides the reader with general information for Hepatocellular carcinoma, miRNA as potential therapeutic agents to combat numerous tumors, and nanostructured lipid carriers.
In the section Materials and methods, the authors have described with details all used methods.
Paragraph 2.3 should be carefully checked for gramma mistakes-line 160.
Section Results
Why are the captions of figures 4 to 8 in boxes?
Paragraph 3.10 in lines 548 and 554 Fig 8A and Fig. 8B are mentioned. However, in the text there is only Fig.8
Line 556 – “**p<0.01” on the figure8 is shown “**p<0.001” – needs correction
Line 647- “PH” – needs correction
I have noticed some mistakes in lines 706, 712, 731.753, 792
The abbreviations PDI and PI (line 728) were used for polydispersity index. The authors should decide on one of them.
Conclusions are logical and derived from the content of the article. The authors also outline directions for future explorations.
The cited literature is up to the point.
I have noticed that one of the author’s names in “Author’s contribution” is El-Agamy. This author is not mentioned in the list after the title. Instead, the name Basma Al Soudy is written. I was wondering whether these names belong to one person?
Comments on the Quality of English LanguageI have noticed typing errors only.
Author Response
Here in attach the response letter and edited manuscript.

Reviewer 4 Report (New Reviewer)
Comments and Suggestions for Authors
The authors developed chitosan-coated nanostructured solid lipid carriers for the delivery of miR-375 through electrostatic interactions. The addition of surfactants made the formulations stable at -20°C freezing conditions. Elaborate in-vitro experiments confirmed increased cellular uptake, reduced cytotoxicity, and effective anti-cancer activities. However, the use of chitosan coating for lipid nanoparticles is widely investigated. What are the strengths of using solid fluid lipid nanoparticles? In the formulation design, the outer layer of lipid nanoparticles was fully covered by chitosan, and miR-375 is mainly attached to the chitosan. I don't understand the functions of solid liquid lipid nanoparticles in this formulation design. It seems that solid liquid lipid nanoparticles only work as an inner core, and any nanoparticles, such as solid lipid nanoparticles or inorganic nanoparticles, could perform the same function. Also, lipid nanoparticles can encapsulate mRNA inside, but this formulation only attaches miR-375 to the surface of nanoparticles. I think this might make miR-375 more vulnerable to the environment (e.g., nuclease) without protection. I also have other comments that need to be addressed before publication:
1. The concentration of Kolliphore P188 used for stability improvement already exceeds the CMC of Kolliphore P188. How can you confirm that the size measured by DLS is not due to the formation of micelles?
2. The storage stability study only checks the physical properties of nanoparticles, but the potency of miR-375 was not evaluated. Also, since the binding of miR-375 to chitosan is not a covalent bond, how much of the miR-375 will dissociate from the formulation during storage?
3. How do you measure the zeta potential? The measured zeta potential is determined by the pH of the solution. The increased pH of the chitosan solution will undoubtedly reduce the zeta potential. It's difficult to tell if the coating of chitosan on nanoparticles has changed or not.
4. The detailed formulation recipe of each group in Table 2 needs to be listed. It's hard for the audience to compare the different groups.
5. The size of nanoparticles after miR-375 loading increased from 65 nm to 200 nm. The significant size and PDI changes might result from aggregation rather than the formation of the miR-375 layer.
6. In in-vitro anti-cancer studies, using doxorubicin at a 200 ng dose is not a good positive control because doxorubicin is a small molecular drug. Why not use TurboFect/miR375 as a positive control group?
7. In figure 6, the bright field images of HepG2 cells are needed, and the scale bar is missing.
8. In lines 750-752, I don't think it represents a significant difference.
Comments on the Quality of English LanguageMinor edition
Author Response
Here in attach the response letter and edited manuscript.

This manuscript is a resubmission of an earlier submission. The following is a list of the peer review reports and author responses from that submission.
Round 1
Reviewer 1 Report
Comments and Suggestions for Authors
The manuscript entitled, ‘Preparation and Optimization of the Anti-Hepatocellular carcinoma Effect of miR-375 Nano-vector using A Novel Nano Structure Solid Lipid Nanoparticles’ suggests the formulation of a solid lipid nano carrier based on solid lipid nanoparticles (Nps) for transportation of miR-375 in Hepatocellular carcinoma cells. Furthermore, the authors have tested the stability and tissue-specific targeting potential of the synthesized material. The concept of the study is sound and the title is interesting. Yet, manuscript lacs of the presentation part and some major concerns should be addressed by the authors prior to any possible consideration of this manuscript to be published in the journal of ‘Pharmaceutics’.
1. The ‘abstract’ is not well written and does not depict the summary of this work clearly. It should be rewritten.
2. The abbreviated words should be explained at their first use in the manuscript. For instance, miRNA (#line_16), HepG2 (#line_24), and MTT (#line_26).
3. The abbreviated words should be consistent throughout the manuscript. Eg., …cationic solid lipid nano carrier (NSLC) (#line_21) or …nanostructured lipid carrier (NSLC) (#line_69). Which one is correct?
4. The statement written in #line_59-61, “Both viral and non-viral delivery systems have been developed to overcome these obstacles but the associated immunogenicity and possible integration into the host genome causing toxicity of the viral-vector based delivery didn’t support its application [15,16]” is confusing and should be rephrased.
5. The authors must arrange the Tables in a serial order in the manuscript. In the manuscript, Table 5 comes after the Table 3, and Table 4 comes in the last. It should be corrected.
6. The authors must prepare suitable #Figures and #Table for a manuscript in a scientific presentation. Except #Table_1,2, and 4, all the tables and figures of the manuscript should be prepared again for the better and clearer presentation of their findings.
7. Figure legends should be comprehensive (#Figure 1, 3, 4, and 5).
8. The location of their construct on the gel (gel pictures) should be highlighted for better comparison and clarification.
9. How the TEM result of ‘blank F2 with 1.5 wt% chitosan’ (#Figure_2B) shows and significant size difference with ‘F2 /miRNA (NSLC containing 1.5 wt% chitosan, 100:1, w/w)’ (#Figure_2D). The authors should explain the possible reason behind it.
10. The author have mentioned, “As shown in Figure 7, the cells treated with F2/miR-375 nano-plex showed greater “green color” intensity compared to treated cells with F1/miR-375 and free miRNA”. The ‘green color’ intensity is due to what? Which dye/stain were used and what was the purpose of the fluorescence microscopy? It should be clearly explained in both ‘Materials and methods’ and ‘Results and discussion’ section.
11. Where is #Figure_8 (results of MTT assay) of the manuscript?
12. In ‘Conclusion’ section the authors have mentioned “These 2 nano-formulations that possess high compatibility……. High compatibility against what? In which experiment the compatibility of the material have been assessed?
13. The manuscript should thoroughly be checked for the English language and grammatical errors.
Reviewer 2 Report
Comments and Suggestions for Authors
This study provides a new therapeutic strategy for HCC through optimized NSLCs preparations. However, the innovation point is not clear, and the verification experiment on the therapeutic effect of miRNA is incomplete. The full text data processing is very rough. There are so many writing mistakes in the paper. Please check and correct them carefully.
Here some points are listed for further correction:
1 Pay attention to the upper and lower case of units and the space between numbers and units in the full text. Please check and correct them carefully. For example, change 0.45Mm to 0.45 mM in 2.2.; There should be a space in 25℃.;0.225 mg/ml or 1500 µl should be rewritten as 0.225 mg/mL and 1500 µL etc.
2. Pay attention to the typography and font of the article.
3. Why is the description of PS,ZP and PDI of F1 nanoparticles in part 3.2 inconsistent with the results in Table2?
4. Note that the formula of MTT in 2.13 is wrong.
5. Please give the reference cited in the miR-375 expression construction and explain what the results 1,2,3,4,5 represent respectively.
6. In Table3, what do miR7-F1, miR34-F1, miR7-F2 and miR34-F2 represent respectively, and what is the purpose of introducing them? The proportion of miR-375 in liposomes is very small. Why does the particle size change so much after loading miR-375?
7. Transmission electron microscopy results of nanoparticles in Figure2 are inconsistent with particle size and potential results of nanoparticles in Table3.
8. Where is the Figure 8 in the article??
9. FTIR spectrum are not very clear, and F1 and F2 are also absorbed at 3400cm-1 and 3200cm-1. Therefore, it is not accurate to prove that the successful combination of miR-375 with F1 and F2 by the absorption of miR375-F1 and miR375-F2 at 3400cm−1 and 3200cm−1.
10. All the images in the article are so poor. Please confirm whether the font in the picture is consistent with the font in the article. Is there a specific purpose for the typography? For example, Figure 2. and Figure 5. Please give more discussion and illustration of Figure 3.
11. The Table is arranged incorrectly. Please check it carefully.
12. To provide the raw data of cellular uptake confocal is necessary, supplemented with flow uptake as an solid evidence.